# Protecting Group-Free Synthesis of Glycopolymer-Type Amphiphilic Macromonomers and Their Use for the Preparation of Carbohydrate-Decorated Polymer Particles

**DOI:** 10.3390/biom9020072

**Published:** 2019-02-19

**Authors:** Jin Motoyanagi, Minh Tan Nguyen, Tomonari Tanaka, Masahiko Minoda

**Affiliations:** 1Faculty of Molecular Chemistry and Engineering, Graduate School of Science and Technology, Kyoto Institute of Technology, Matsugasaki, Sakyo-ku, Kyoto 606-8585, Japan; jinmoto@kit.ac.jp (J.M.); nmtanmtx@gmail.com (M.T.N.); 2Department of Biobased Materials Science, Graduate School of Science and Technology, Kyoto Institute of Technology, Matsugasaki, Sakyo-ku, Kyoto 606-8585, Japan; t-tanaka@kit.ac.jp

**Keywords:** carbohydrate-decorated polymer particle, glycopolymer-type macromonomer, living cationic polymerization, click reaction, poly(vinyl ether)

## Abstract

Polymer particles modified with carbohydrates on their surfaces are of significant interest, because their specific recognition abilities to biomolecules are valuable for developing promising materials in biomedical fields. Carbohydrate-decorated core-shell polymer particles are expected to be efficiently prepared by dispersion polymerization using a glycopolymer-based amphiphilic macromonomer as both a polymeric steric stabilizer and a monomer. To create glycopolymer-type macromonomers, we propose a new strategy combining living cationic polymerization of an alkynyl-functionalized vinyl ether (VE), and the click reaction for the preparation of glycopolymers having a polymerizable terminal group, and investigate their dispersion copolymerization with styrene for generating carbohydrate-decorated polymer particles. This study deals with (i) the synthesis of block copolymer-type amphiphilic macromonomers bearing a methacryloyl group at the *α*-terminus, and pendant alkynyl groups by living cationic polymerization of alkynyl-substituted VE (VEEP), (ii) the derivatization of maltose-carrying macromonomers by click chemistry of the pendant alkynyl groups of the precursor macromonomers with maltosyl azide without any protecting/deprotecting processes, and (iii) the preparation of maltose-decorated (Mal-decorated) polymer particles through the dispersion copolymerization of glycopolymer-type macromonomers with styrene in polar media. Moreover, this study concerns the specific interactions of the resultant polymer particles with the lectin concanavalin A (Con A).

## 1. Introduction

Surface-functionalized polymer particles have attracted increased attention because of their scientific and technical importance [1,2,3] as versatile materials with extremely large specific surface areas. In particular, polymer particles decorated with carbohydrates on their surfaces have been of great interest in the biomedical field, because they have the potential ability for providing useful materials for diagnosis based on their specific interactions with biomolecules, such as proteins and viruses, etc. [4,5,6,7]. One effective method for producing surface-functionalized particles is the use of amphiphilic macromonomers for dispersion or emulsion copolymerization with hydrophobic monomers in polar media [8,9,10]. The resultant polymer particles afford stable aqueous dispersions, where the dispersion stability is attributed to the steric stabilization attained by hydrophilic polymer chains fixed on the particle surfaces. It is worth noting that the surface functionality of the particles can be widely designed by varying the structure of the macromonomers because they are covalently attached to the particle surfaces. We have already reported that poly(vinyl ether) (polyVE)-based hydrophilic macromonomers bearing functional side chains can produce monodispersed polymer particles, where the surfaces of the polymer particles can be modified with a variety of functions originating from the pendant functions of the macromonomers employed [11,12]. In contrast, there is still little information available on the use of glycopolymer-type macromonomers for the formation of core-shell polymer particles [13,14], therefore, synthesis of such macromonomers having glycopolymer backbones, and studies on their potential ability as steric stabilizers (glycostabilizers) for forming polymer particles are strongly required. The synthesis of well-defined glycopolymers have been achieved by the controlled polymerization of carbohydrate-containing monomers, or the post-functionalization of the precursor reactive polymers using carbohydrate-based agents as exemplified in copper(I)-catalyzed alkyne-azide cycloaddition (CuAAC) [15,16]. A large number of papers have reported the synthesis of well-defined glycopolymers [17] by various controlled polymerizations such as living radical polymerization [18,19,20], living ionic polymerization [21,22,23], and ring-opening metathesis polymerization [24]. An alternative approach to the synthesis of well-defined glycopolymers is a CuAAC-based procedure, in which alkyne-carrying precursor polymers are precisely synthesized and then decorated with azide-functionalized carbohydrates by click reaction.

In this study, for the preparation of carbohydrate-decorated polymer particles, we propose a synthetic strategy for designing novel glycostabilizers that correspond to the glycopolymer-type macromonomers that possess both a terminal polymerizable group and a carbohydrate-substituted amphiphilic block copolymer backbone (Figure 1). The living cationic sequential block copolymerization of isobutyl vinyl ether (IBVE) and an alkyne-functionalized VE (VEEP) using the initiator bearing a methacryloyl group (VEM-HCl) produced block copolymer-type precursor macromonomers (MA-PIBVE-*b*-PVEEP) bearing a methacryloyl group at the *α*-terminus. This precursor macromonomer was then subjected to the CuAAC click reaction with maltosyl azide to form the target amphiphilic glycopolymer-type macromonomer (MA-PIBVE-*b*-P(VE-Mal)) The maltose-decorated (Mal-decorated) polymer particles were then prepared by dispersion copolymerization of the obtained macromonomer and styrene, and their lectin-binding properties were also investigated.

## 2. Materials and Methods

### 2.1. Chemicals and Reagents

d(+)-maltose monohydrate, sodium azide, 2-chloro-1, 3-dimethylimidazolinium chloride (DMC), copper (II) sulfate pentahydrate, l-ascorbic acid sodium salt, 2,2′-azobis-[2-(2-imidazolin-2-yl)-propane] dihydrochlroride (VA-044), styrene, concanavalin A (Con A) from *Canavalia ensiformis*, and bovine serum albumin (BSA) were purchased from FUJIFILM Wako Pure Chemical Corporation (Osaka, Japan). *N*,*N*-diisopropyl ethylamine (DIPEA) and tris[(1-benzyl-1*H*-1,2,3-triazol-4-yl)methyl]amine (TBTA) were purchased from Tokyo Chemical Industry Co., Ltd. (Tokyo, Japan). Isobutyl vinyl ether (IBVE) was purchased from Sigma-Aldrich (St. Louis, MO, USA). Cellulose dialysis tubes size 20 with molecular weight cut-off MWCO = 14 kDa and pre-wetted dialysis tubing with MWCO = 1 kDa and 2 kDa (nominal flat width: 18 mm, diameter 11.5 mm, volume: 1.1 mL/cm, length: 10 m) (Spectrum, Rancho Dominguez, CA, USA) was performed in a 1 L beaker by changing the distilled water four or five times over a period of 24 h. Anthrone reagent was prepared by adding 200 mg anthrone into 100 mL ice-cold 95% H_2_SO_4_. All other reagents were commercially available, and used without further purification. Synthesis of 2-(vinyloxy)ethyl methacrylate (VEM) and the VEM-HCl adduct were carried out according to the procedure reported in the literature [25]. Maltosyl azide was prepared according to the literature [26,27]. Synthesis of 3-[2-(2-vinyloxyethoxy)-ethoxy]-propyne (VEEP) was carried out by a reaction of 2-(vinyloxyethoxy)-ethanol with propargyl bromide in the presence of KOH in dimethyl sulfoxide (DMSO) at room temperature for 40 h [28].

### 2.2. Methods

Nuclear magnetic resonance (NMR) spectra of ^1^H and ^13^C were recorded at 25 °C on a Bruker model AC-500 spectrometer, operating at 500 and 125 MHz, respectively, where chemical shifts (*δ* in ppm) were determined with respect to non-deuterated solvent residues as internal standards. Preparative size exclusion chromatography (SEC) was performed at 25 °C by using 21.5 mm × 300 mm polystyrene gel columns (TOSOH TSKgel G2000H, G2500H and G3000H) on a TOSOH model CCPE equipped with an RI-8022 RI detector. Analytical SEC was performed in tetrahydrofuran (THF) at 40 °C, using 8.0 mm × 300 mm polystyrene gel columns (Shodex KF-804 × 2) on a TOSOH model DP-8020 equipped with a UV-8000 variable-wavelength UV–vis detector and a RI-8022 RI detector. The number-average molecular weight (*M*_n_) and polydispersity ratio (*M*_w_/*M*_n_) were calculated from the chromatographs with respect to 15 polystyrene standards (Scientific Polymer Products, Inc., Ontario, NY, USA; *M*_n_ = 580–670,000 g/mol, *M*_w_/*M*_n_ = 1.01−1.07). Analytical SEC was performed in 0.2 mol/L NaNO_3_ aqueous solution at 40 °C, using 7.8 mm × 300 mm gel columns (TOSOH TSKgel *α* −3000 × 3) on a JASCO model PU2089 equipped with a UV-2075 variable-wavelength ultraviolet–visible (UV–Vis) detector and an RI-2031 RI detector. The number-average molecular weight (*M*_n_) and the polydispersity ratio (*M*_w_/*M*_n_) were calculated from the chromatographs with respect to poly(ethylene glycol)s standards (Scientific Polymer Products, Inc., Ontario, NY, USA; *M*_n_ = 590–11,900 g/mol, *M*_w_/*M*_n_ = 1.05−1.11). Scanning electron microscopy (SEM) images were observed by using a JEOL JSM-7600F. UV–Vis spectra were recorded on a SHIMADZU Type UV-2550 spectrometer.

### 2.3. Synthesis of MA-PIBVE-b-PVEEP

Synthesis of the precursor block macromonomer (MA-PIBVE-*b*-PVEEP) having both a terminal methacryloyl group and pendant alkynyl functions was carried out by the living cationic polymerization of IBVE (600 mmol/L, 1.35 mL in toluene) and VEEP (600 mmol/L, 1.35 mL in toluene) in this order, with a prechilled solution of VEM-HCl (20 mmol/L, 0.45 mL in toluene) as an initiator, and ZnI_2_ solution (2.0 mmol/L, 0.45 mL in toluene) as an activator at −30 °C under a dry nitrogen atmosphere in a baked glass tube equipped with a three-way stopcock. Toluene (0.9 mL) was added to the solution mixture to make up to 4.5 mL in total volume. After a period of 10 h, the polymerization was quenched with an excess of amount of chilled ammonia solution in methanol (CH_3_OH/NH_3_ aq.). The obtained reaction mixture was diluted with toluene and washed with 20% aqueous sodium thiosulfate solution and water to remove the salts, evaporated to dryness under reduced pressure, then vacuum-dried to yield the precursor macromonomer (MA-PIBVE-*b*-PVEEP) (Products P1 and P2). The *M*_n_ and *M*_w_/*M*_n_ of MA-PIBVE-*b*-PVEEP were estimated by analytical SEC in THF. Furthermore, the structural analysis of the target product was performed by ^1^H NMR spectroscopy in CDCl_3_.

### 2.4. Synthesis of MA-PIBVE-b-P(VE-Mal)

Precursor macromonomers MA-PIBVE-*b*-PVEEP and maltosyl azide (Mal-N_3_) were suspended in a 1:1 mixture of THF and water (20 mL). Sodium ascorbate (20 mol%) was added, followed by the addition of copper (II) sulfate pentahydrate (10 mol%) and TBTA (10 mol%). The heterogeneous mixture was stirred vigorously for 92 h under nitrogen atmosphere at room temperature, at which point it became transparent, and TLC analysis indicated the complete consumption of the reactants. The reaction mixture was extracted with toluene and evaporated to dryness under reduced pressure. The water solution of the reaction mixture was poured into a large amount of methanol to precipitate the polymers. The resultant polymer was collected by centrifugation, and dried under reduced pressure. For further purification, the crude product was purified by dialysis membrane with a molecular weight cut-off (MWCO) of 2000 in distilled water for more than two weeks, and then recovered by freeze-drying. The products (P3 and P4) were off-white colored solids.

### 2.5. Preparation of Mal-Decorated Polymer Particles

The obtained block macromonomers MA-PIBVE-*b*-P(VE-Mal) (5.0 µmol), styrene (520 mg, 5 mmol), and VA-044 (16 mg, 50 µmol) were dissolved in the mixture solvent (10 mL, EtOH/H_2_O = 4/1, v/v) in a glass tube. The solution was degassed by several freeze–thaw cycles, then the glass tube was sealed off and shaken (150 shakes per minute) at 50 °C for 5h. The products were purified by dialysis (MWCO 14,000) against deionized water, and freeze-dried to give polymer particles. The morphologies of the resultant polymer particles were examined by SEM analysis.

### 2.6. Lectin Binding Assay

The protein (Con A or BSA; 0.1 mL, 300 μM) was added to the polymer particle suspension (0.1 g/L) in the buffer solution (4.0 mL, 0.1 M Tris-HCl, 1 mM MnCl_2_, 1 mM CaCl_2_, 10 mM NaCl, pH 7.5). The transmittance of the supernatant was monitored at 550 nm after 2 h.

### 2.7. Determination of Maltose Density on Polymer Particles

Polymer particles (10.0 mg) were heated at 80 °C under acid conditions (2.5 M HCl aq, 5 mL) for 3 h. The resultant solution was neutralized by sodium carbonate. The supernatant was collected, and anthrone solution was added (200 mg anthrone and 100 mL H_2_SO_4_). The mixture was heated at 80 °C for 10 min, and then cooled rapidly. The absorbance at 630 nm was measured by UV–Vis spectrometry. The standard curve was made by using a glucose solution.

## 3. Results and Discussion

### 3.1. Synthesis of MA-PIBVE-b-PVEEP

The precursor block macromonomer (MA-PIBVE-*b*-PVEEP) possessing a methacryloyl group at the *α*-terminus, and pendant alkynyl functions, was synthesized by living cationic sequential block copolymerization of IBVE and VEEP using the VEM-HCl adduct/ZnI_2_ initiating system [11,12,25]. ([IBVE]_0_/[VEEP]_0_/[VEM-HCl]_0_/[ZnI_2_]_0_ = 30/30/1/0.1, [IBVE]_0_ and [VEEP]_0_ = 600 mmol/L). After quenching the polymerization, the product was purified by preparative SEC to remove the unreacted IBVE and VEEP. The product was soluble in common organic solvents. Table 1 shows the results of SEC and ^1^H NMR analysis for the isolated block copolymers. As shown in Figure 2a, the analytical SEC data of the isolated block macromonomer depicts a unimodal MWD curve without any shoulders, as well as the narrower molecular weight distribution (*M*_w_/*M*_n_ < 1.63) compared to those obtained by conventional cationic polymerization. These results indicate that neither the methacryloyl moiety at the α-end, nor the pendant alkynes, inhibit living cationic polymerization. The formation of the target block macromonomer was confirmed by ^1^H NMR measurements in CDCl_3_ (Figure 3a). All of the key signals arising from the methacryloyl group at the *α*-terminus (5.55 and 6.12 ppm), and the IBVE repeating units (0.9, 3.1–3.3 ppm) and VEEP repeating units (2.5, 3.5–3.67 and 4.19 ppm) were consistent with those of the expected structure for the MA-PIBVE-*b*-PVEEP. The number-average degree of polymerization (*DP*_n_) of MA-PIBVE-*b*-PVEEP was determined by the peak intensity ratio of the isobutyl protons (peak c) and the alkynyl protons (peak a), based on the terminal vinyl protons (peak e) for the IBVE and VEEP segments, respectively. Furthermore, the ^13^C NMR spectrum in Figure 2b clearly showed the carbon peaks that were assignable to the pendant alkynyl groups (79.57 and 80.68 ppm) and the dimethyl groups of IBVE (28.78 and 19.44 ppm), respectively. However, the vinyl and carbonyl carbons of the terminal methacryloyl moiety were difficult to observe, probably due to their much lower concentrations compared to the pendant alkynyl and methyl carbons. In addition, as for the carbonyl carbon, no appearance in the spectra might also be caused by its inherent lower intensity than other carbons, in ^13^C NMR spectroscopy. These results indicate that the macromonomer MA-PIBVE-*b*-PVEEP, having both a terminal methacryloyl group and an alkynyl groups-substituted block copolymer backbone, was precisely synthesized by living cationic polymerization.

### 3.2. Synthesis of MA-PIBVE-b-P(VE-Mal)

We investigated the conversion of the pendant alkynyl groups of MA-PIBVE-*b*-PVEEP to carbohydrate residues by the CuAAC click reaction, which was carried out in THF/H_2_O (1/1 (*v*/*v*)) at 30 °C under dry nitrogen atmosphere for 92 h in the presence of CuSO_4_ and sodium ascorbate as a catalyst and a reductant, respectively (Table 1). In this study, a small excess amount of Mal-N_3_ was reacted with the pendant alkynyl moieties ([VEEP**^*^**]_0_/[Mal-N_3_]_0_/[CuSO_4_]_0_/[sodium ascorbate]_0_/[TBTA]_0_ = 1/1.01/0.1/0.2/0.1) ([VEEP^*^] shows the calculated molar concentration of the VEEP repeating units). After the reaction mixture was washed with toluene to remove the TBTA, the product was purified by reprecipitation and dialysis to remove the unreacted Mal-N_3_ and traces of the copper catalyst. The isolated yields were relatively low, probably due to losses in the purification processes, because we thoroughly purified the product by combining reprecipitation and dialysis over a couple of weeks, because the high purity of the macromonomer was indispensable for the success of the post-reactions. Consequently, the SEC trace in Figure 2c indicates the successful purification of the maltose-substituted glycopolymer-type macromonomer. The occurrence of the CuAAC click reaction with Mal-N_3_ led to a drastic change in the solubility characteristics of the macromonomers. As a result, the obtained maltose-appended macromonomers (P3 and P4) were readily soluble in a mixture of EtOH and water, while the precursor macromonomers (MA-PIBVE-*b*-PVEEP, P1 and P2) were sparingly soluble in EtOH and insoluble in water. The isolated polymer was subjected to ^1^H NMR analysis. As shown in Figure 3b, multiple signals at 5.9–3.1 ppm, a sharp signal at 0.83 ppm, and broadened signals at 1.9–1.1 ppm were observed, which were assignable to the pendant maltose moieties, the dimethyl group of the IBVE segment, and the polyVE main chain moiety, respectively. In addition, strong key signals appeared at 8.3 ppm and 4.5 ppm. These are ascribable to the linkage triazole proton (peak a’) and methylene protons adjacent to the triazole moiety (peak b’), respectively. Based on the peak intensity ratio of the triazole proton to the methylene protons adjacent to the residual alkynyl group (4.2 ppm), after click reaction, more than 95% of the pendant alkynes were converted to the triazole units, indicating that the click reaction of the pendant alkynes of MA-PIBVE-*b*-PVEEP and maltosyl azide nearly quantitatively proceeded. The proton concentrations of the terminal methacryloyl moiety is quite low; however, the ^1^H NMR spectrum showed a signal arising from a vinyl proton of the terminal methacryloyl group at around 6.0 ppm (peak e’) (Figure 3b, inset). These results confirm that glycopolymer-type amphiphilic macromonomers MA-PIBVE-*b*-P(VE-Mal) having both a terminal methacryloyl group and pendant maltose moieties was precisely synthesized by a combination of living cationic polymerization and CuAAC click reaction.

### 3.3. Preparation of Mal-Decorated Polymer Particles

To prepare Mal-decorated polymer particles, the obtained glycopolymer-type amphiphilic macromonomers [MA-PIBVE-*b*-P(VE-Mal)] were subjected to dispersion copolymerization with styrene. In this study, two kinds of MA-PIBVE-*b*-P(VE-Mal) having different *DP*_n_ values of P(VE-Mal) segment (P3; *DP*_n_ = 10 and P4; *DP*_n_ = 30, respectively) were synthesized and employed. Dispersion polymerization was carried out in EtOH/water (4/1 (*v*/*v*)), with VA-044 as the initiator at 50 °C, and at over 150 shakes per minute, for 5 h. After the dialysis of the reaction mixture for the removal of the unreacted amphiphilic macromonomer, followed by lyophilization, the polymerization product was obtained as a white powder. Scanning electron microscope (SEM) of an air-dried sample from the suspension in EtOH indicated the presence of submicron-sized uniform and spherical polymer particles (Table 2 and Figure 4). It should be noted that the size of the polymer particles was dependent on the *DP*_n_ of the P(VE-Mal) segment of the macromonomer: the higher the *DP*_n_ of the P(VE-Mal) segment, the larger the particle diameter. This result indicates that the size of the polymer particles can be controlled by varying the *DP*_n_ of the glycopolymer-type macromonomer. The glycopolymer-based amphiphilic macromonomer with a hydrophilic P(VE-Mal) segment, and a hydrophobic PIBVE segment thus effectively worked as a surfactant, allowing for the preparation of styrene-based core-shell polymer particles by dispersion polymerization in aqueous solvent. Furthermore, stable dispersions of the Mal-decorated polymer particles can be obtained, not only in alcoholic solvents such as a mixture EtOH and water but also in buffer solutions.

### 3.4. Lectin Binding Assay and Estimation of Maltose Density

It has been reported that lectin Con A specifically interacts with the *α*-glucoside residue of a maltose moiety. Thus, we investigated the interaction of the Mal-decorated particles with Con A. Some differences in the results of lectin binding assay were observed to be dependent on the *DP*_n_ of the P(VE-Mal) segment of the macromonomer employed. As for P3, no significant differences in transmittance were observed for the three aqueous suspensions (control, with Con A, and with BSA) (Figure 5a), indicating Mal-decorated polymer particles prepared from the glycopolymer-type macromonomer P3 with a shorter P(VE-Mal) segment did not strongly interact with Con A. In sharp contrast, the transmittance of the aqueous suspension of the polymer particle prepared from P4 with a longer P(VE-Mal) segment increased remarkably upon the addition of Con A; this was due to the precipitation of the lectin-particle conjugates (Figure 5b). Compared to the result of the addition of Con A, polymer particles prepared with P4 showed no increase in transmittance upon the addition of BSA as a control protein. These results support that the particles prepared using P4 more strongly bound to Con A than those prepared with P3. The strength of the interaction of the polymer particles with Con A is presumably dependent on the carbohydrate density on the particle surfaces. Then, the density of maltose moieties on the particle surfaces was quantitatively evaluated by measuring the amount of the liberated glucose moieties that are released from the polymer particle surfaces by hydrolysis under acidic conditions. This assay showed that the surface densities of maltose moieties on the particles prepared using P4 and P3 were 3.30 and 1.90 µg/cm^2^, respectively (Table 3). These results suggest that the higher density of the maltose residues on the particle surfaces brought about the higher extent of interaction with Con A (glycocluster effect) [29,30]. Here, it should be emphasized that the saccharide density of the polymer particle surfaces can be regulated by the hydrophilic/hydrophobic balance and the *DP*_n_ of the block copolymer-type glycostabilizer.

## 4. Conclusions

We have succeeded in the synthesis of new glycopolymer-type macromonomers (MA-PIBVE-*b*-P(VE-Mal)), consisting of an amphiphilic block copolymer backbone bearing maltose moieties and a vinyl group at the *α*-terminus, by a combination of living cationic sequential block copolymerization of IBVE and VEEP, and a CuAAC click reaction with maltosyl azide, without any protecting/deprotecting processes. We have also demonstrated the preparation of nearly monodispersed Mal-decorated polymer particles by dispersion copolymerization with styrene, using the glycopolymer-type macromonomers as steric stabilizers. We further confirmed the successful formation of the Mal-decorated polymer particles, and their capability for specific binding to Con A by the lectin binding assay, employing BSA as a control protein. Glycopolymer-type macromonomers with controlled architectures are expected to make a notable contribution to the development of carbohydrate-based functional polymeric materials.

## Figures and Tables

**Figure 1 biomolecules-09-00072-f001:**
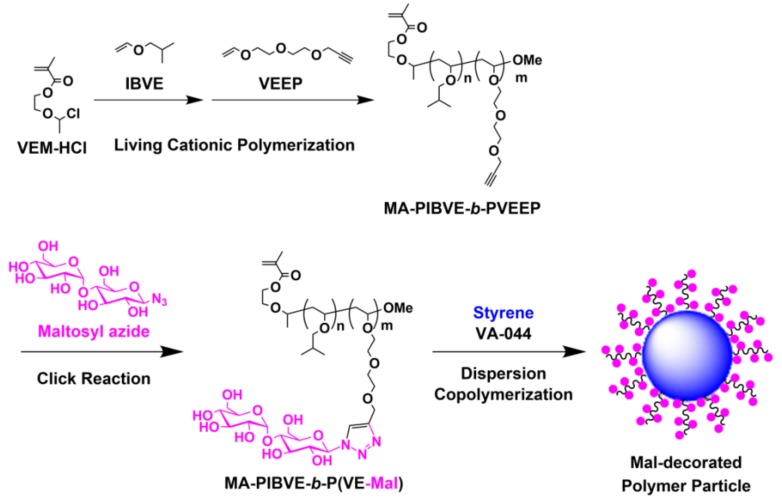
Synthesis of glycopolymer-type amphiphilic macromonomers (MA-PIBVE-*b*-P(VE-Mal)) and their application to the preparation of maltose-decorated (Mal-decorated) polymer particles by dispersion polymerization.

**Figure 2 biomolecules-09-00072-f002:**
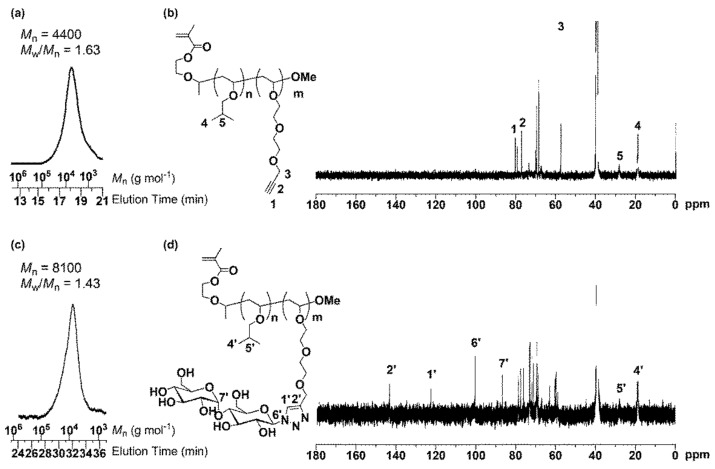
(**a**) Size exclusion chromatography (SEC) curve of MA-PIBVE_25_-*b*-PVEEP_10_ using tetrahydrofuran (THF) as the eluent. (**b**) ^13^C Nuclear magnetic resonance (NMR) spectrum of the MA-PIBVE_25_-*b*-PVEEP_10_ macromonomer in CDCl_3_. (**c**) SEC curve of MA-PIBVE_25_-*b*-P(VE-Mal)_10_ using 0.2 mol/L NaNO_3_ (aq.) as the eluent. (**d**) ^13^C NMR spectrum of MA-PIBVE_25_-*b*-P(VE-Mal)_10_ macromonomer in DMSO-*d*_6_.

**Figure 3 biomolecules-09-00072-f003:**
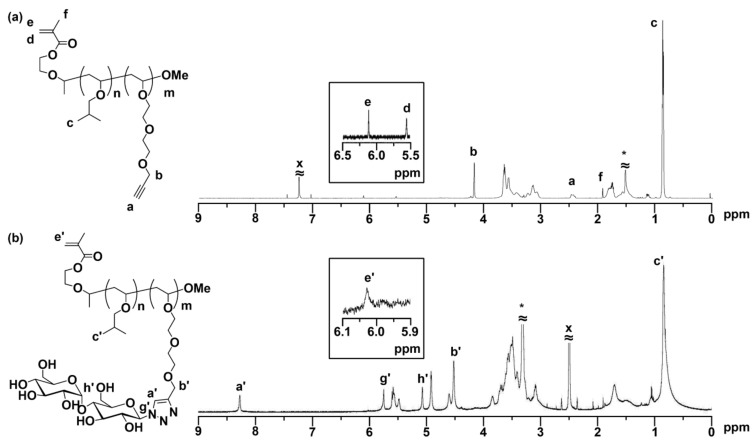
^1^H nuclear magnetic resonance (NMR) spectra of (**a**) MA-PIBVE_25_-*b*-PVEEP_10_ in CDCl_3_ and (**b**) MA-PIBVE_25_-*b*-P(VE-Mal)_10_ in DMSO-*d*_6_ (× and *; remaining solvents).

**Figure 4 biomolecules-09-00072-f004:**
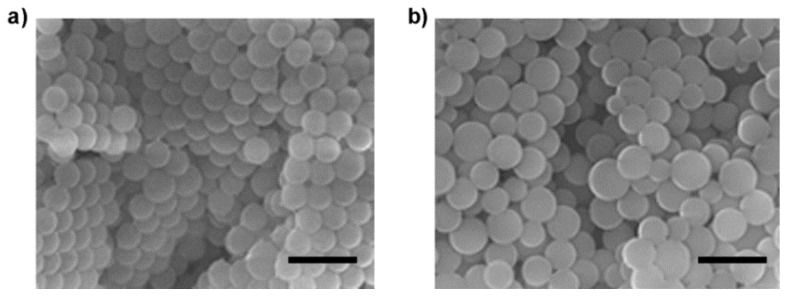
SEM images of the Mal-decorated polymer particles obtained by dispersion copolymerization employing (**a**) P3 and (**b**) P4 of MA-PIBVE-*b*-P(VE-Mal). Bars: 1 μm.

**Figure 5 biomolecules-09-00072-f005:**
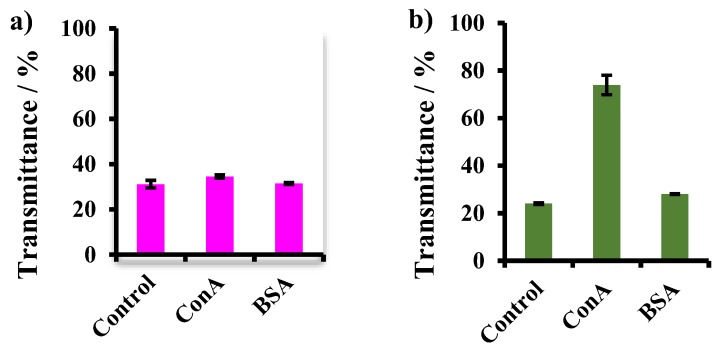
Transmittance of Mal-decorated polymer particle suspensions prepared using (**a**) P3 and (**b**) P4 after adding lectins for 2h. Control: polymer particle suspension before the addition of Con A and BSA.

**Table 1 biomolecules-09-00072-t001:** Synthesis of MA-PIBVE-*b*-P(VE-Mal).

Entry	Precursor Macromonomer					CuAAC Click Reaction Product				
		$M_{n}^{1}$	$M_{w}/M_{n}^{2}$	*DP*_n_ of PIBVE ^1^	*DP*_n_ of PVEEP ^1^		$M_{n}^{1}$	$M_{w}/M_{n}^{3}$	Yield (%) ^4^	Degree of Substitution (%) ^1^
1	P1	4400	1.63	25	10	P3	8100	1.43	20	Quant.
2	P2	7800	1.60	25	30	P4	18,800	2.13	15	Quant.

^1^ Determined by ^1^H nuclear magnetic resonance (NMR) spectrum. ^2^ Estimated by polystyrene-calibrated size exclusion chromatography (SEC). ^3^ Estimated by poly(ethylene glycol)-calibrated SEC. ^4^ Isolated yield.

**Table 2 biomolecules-09-00072-t002:** Preparation of Mal-decorated polymer particles.

Particle	Yield (%) ^1^	*D*_n_ (nm) ^2^	PDI ^2^
P3-Particle	55	460 ± 80	1.19
P4-Particle	55	550 ± 110	1.24

^1^ Isolated yield. ^2^ Determined by SEM analysis (average of 100 particles).

**Table 3 biomolecules-09-00072-t003:** Determination of maltose amounts on the Mal-decorated polymer particle surfaces.

Particle	Absorbance at 630 nm ^1^	Concentration of Free Glucose (μg/mL)	Amount of Free Glucose (mg/8.1 mg)	Amount of Maltose (mg/8.1 mg)	*D*_n_ (nm) ^2^	Surface Area of the Polymer Particle (cm^2^) × 10^−10^	Surface Density of Maltose Moieties on the Particles (µg/cm^2^)
P3-Particle	0.328	23.8	0.298	0.595	480	18.0	1.90
P4-Particle	0.419	35.6	0.447	0.894	550	24.1	3.30

^1^ Estimated by polystyrene-calibrated SEC. ^2^ Determined by SEM analysis (Average of 100 particles).

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
