# Peer review of "Protecting Group-Free Synthesis of Glycopolymer-Type Amphiphilic Macromonomers and Their Use for the Preparation of Carbohydrate-Decorated Polymer Particles"

_biomolecules, 2019, doi:10.3390/biom9020072_

Round 1
Reviewer 1 Report
The manuscript “Protecting Group‐Free Synthesis of Glycopolymer‐Type Amphiphilic Macromonomers and their Use for the Preparation of Carbohydrate‐Decorated Polymer Particles” of Jin Motoyanagi, Minh Tan Nguyen, Tomonari Tanaka , and Masahiko Minoda describes the development of surface‐functionalized polymer particles. The topic is very current and the research group has been involved for some time in this field. The methodologies used in this study are appropriate and the results are well discussed. I am not absolutely sure that the article is in line with the topics covered by the journal, being a research focused on synthesis without any applicative studies of the particles.
According to me, the manuscript needs some modifications.
In particular:
Page 3, line 80. Correct Chemicala
Page 3, line 112 Analytical SEC of ..of what?
The measurement unit is not uniform (ml or mL?)
Page 4, line 146. Distilled water at pH = 7. Are you sure of this value? It seems unlikely.
Page 5, Table 1. The yield of P3 and P4 is very low. Have the authors tried to explain why?
The quality of Figure 1 is not excellent and should be improved.
In the 13C-NMR spectrum the carbonyl signal of the methacrylic group is not visible. The authors should justify the reason.
Page 6. Preparation of Mal-decorated polymer particles. The authors recently published an article
on these particles (Chemistry Letters, 2018, 47: 1519-1521), which should be reported in the
references.
Author Response
Answers to the Comments of Reviewer 1;
We appreciate your valuable comments and suggestions. With careful consideration of your comments, we have modified the manuscript and answered to the questions as follows.
Page 3, line 80. Correct Chemicala
--> We have revised the subhead as follows:
Page 3, lines 80;
“2.1. Chemicals and Reagents”
Page 3, line 112 Analytical SEC of ..of what?
--> We have revised the sentences as follows:
Page 3, line 110;
“Analytical SEC was performed”
The measurement unit is not uniform (ml or mL?)
--> According to your suggestion, we have revised the unit from “ml” to “mL” in Page 2, line 90 and in Table 3
Page 4, line 146. Distilled water at pH = 7. Are you sure of this value? It seems unlikely.
--> According to your suggestion, we have revised the sentence as follows:
Page 4, line 144;
“… by dialysis membrane with a molecular weight cut-off (MWCO) of 2000 in distilled water for more than… ”
Page 5, Table 1. The yield of P3 and P4 is very low. Have the authors tried to explain why?
--> As you pointed out, the yields of the polymers were relatively low (15-20%). This is probably due to loss in purification processes because we have thoroughly purified the product by combining reprecipitation and dialysis over a couple of weeks because the high purity of the macromonomer is indispensable for the successful post reactions. According to your suggestion, we have added the sentence as follows;
“After the reaction mixture was washed with toluene to remove the TBTA, the product was purified by reprecipitation and dialysis to remove the unreacted Mal-N3 and trace of the copper catalyst. The isolated yields were relatively low probably due to loss in the purification processes because we have thoroughly purified the product by combining reprecipitation and dialysis over a couple of weeks because the high purity of the macromonomer is indispensable for the successful post reactions. Consequently, the SEC trace in Figure 2c indicates the successful purification of the maltose-substituted glycopolymer-type macromonomer.”
The quality of Figure 1 is not excellent and should be improved.
--> According to your suggestion, we have readjusted component objects and refined the chemical structures and schematic illustration of the polymer particle in Figure 1.
In the 13C-NMR spectrum the carbonyl signal of the methacrylic group is not visible. The authors should justify the reason.
--> As you pointed, the vinyl and carbonyl carbons were not observed in the 13C NMR spectra of the precursor and CuAAC click reaction product; this is probably due to much lower concentration of those carbons derived from the terminal methacryloyl moiety compared to those in the pendants. Considering your comment, we have added the sentence as follows;
“However, the vinyl and carbonyl carbons of the terminal methacryloyl moiety were difficult to observe probably due to their much lower concentrations compared to the pendant alkynyl and methyl carbons. In addition, as for the carbonyl carbon, no appearance in the spectra might also be caused by the inherent lower intensity than other carbons in 13C NMR spectroscopy. ”
Page 6. Preparation of Mal-decorated polymer particles. The authors recently published an article on these particles (Chemistry Letters, 2018, 47: 1519-1521), which should be reported in the references.
--> Thank you for your pertinent suggestion. We added the reference in the revised manuscript.
As ref. 5b; Tanaka, T.; Nguyen, M.T.; Minoda, M. Amphiphilic Glycopolymer-type Macromonomers for the Preparation of Carbohydrate-decorated Polymer Particles. Chem. Lett. 2018, 47, 1519-1521.
Reviewer 2 Report
This manuscript demonstrated preparation of a glycopolymer-type anphphilic macromonomer without any protecting groups and a maltose-decorated polymer particle based on the dispersion copolymerization of the macromonomer with styrene. Such a preparation strategy would be useful for development of various functional materials containing glycopolymers. However, the reviewer strongly recommends to take into following comments account before publication in Biomolecules.
(1) Lines 174-178: Authors claimed that no side-reaction of the methacrylaoyl moiety at the a-end and the pendant alkynes occurred during polymerization based on only GPC results. The reviewer thinks that should be carefully discussed based on in conjunction with NMR analysis. Also, we cannot judge the MWD curves have any shoulders or not because the baselines of the curves are unclear. The reviewer encourages the authors to display the whole shape of curves.
(2) Table 1. Why was the yield of P3 and P4 so low? The authors didn’t mention about the reason. It would be better to add some explanations.
(3) Looking at the SEM images, the diameter for the Mal-decorated polymer particles with P3 seems to be smaller than 480 nm. Is this correct?
(4) Lines 234-235 & Table 2: We cannot judge the difference in the size of the particles is significance or not. How about add the standard deviations?
(5) Why there is no results for control experiments using P1 or P2 for preparation of particles and lectin binding assay? The reviewer thinks that is quite important to support the claims in this manuscript.
(6) Heating at 80 ℃ under acidic conditions for polymer particles for 3 h is a quite tough situation for polymers. The reviewer thinks that the poly(vinyl ether) components, especially near the acetal moieties, may be decomposed. In that case, the color of solutions may be changed, which may affect the absorbance values. Did the authors consider that possibility?
Author Response
Answers to the Comments of Reviewer 2;
We appreciate your valuable comments and suggestions. With careful consideration of your comments, we have modified the manuscript and answered to the questions as follows.
(1) Lines 174-178: Authors claimed that no side-reaction of the methacryloyl moiety at the a-end and the pendant alkynes occurred during polymerization based on only GPC results. The reviewer thinks that should be carefully discussed based on in conjunction with NMR analysis. Also, we cannot judge the MWD curves have any shoulders or not because the baselines of the curves are unclear. The reviewer encourages the authors to display the whole shape of curves.
--> As you pointed out, the unimodal GPC curve without any shoulders, as well as narrower molecular weight distribution, is not enough to conclude that no side-reaction of the methacryloyl moiety at the a-end and the pendant alkynes occurred during polymerization. As you mentioned, successful formation of the target macromonomers, meaning absence of any side-reactions of the methacryloyl group at a-end and the pendant alkynes, should be confirmed based on the 1H and 13C NMR data (lines 178-187 in the original manuscript). We have therefore described the conclusion in the last sentence of this paragraph (lines 188-191). Considering your comments, we have revised the sentence in lines 176-178 as follows;
“These results indicate that neither the methacryloyl moiety at the α-end nor the pendant alkynes inhibit living cationic polymerization.”
(lines 174-175 in the revised manuscript)
In addition, according to your suggestion, we have replaced the GPC charts in Figure 2a and 2c with revised ones including whole shape of curves.
(2) Table 1. Why was the yield of P3 and P4 so low? The authors didn’t mention about the reason. It would be better to add some explanations.
--> As you pointed out, the yields of the polymers were relatively low (15-20%). This is probably due to loss in purification processes because we have thoroughly purified the product by combining reprecipitation and dialysis over a couple of weeks because the high purity of the macromonomer is indispensable for the successful post reactions. According to your suggestion, we have added the sentence as follows;
“After the reaction mixture was washed with toluene to remove the TBTA, the product was purified by reprecipitation and dialysis to remove the unreacted Mal-N3 and trace of the copper catalyst. The isolated yields were relatively low probably due to loss in the purification processes because we have thoroughly purified the product by combining reprecipitation and dialysis over a couple of weeks because the high purity of the macromonomer is indispensable for the successful post reactions. Consequently, the SEC trace in Figure 2c indicates the successful purification of the maltose-substituted glycopolymer-type macromonomer.”
(3) Looking at the SEM images, the diameter for the Mal-decorated polymer particles with P3 seems to be smaller than 480 nm. Is this correct?
--> According to your suggestion, we have estimated the average diameter of polymer particles again, where we have randomly picked up 100 particles then measured their diameters. The re-estimated value of average diameter was 460 nm. We revised the average diameter in Table 2.
(4) Lines 234-235 & Table 2: We cannot judge the difference in the size of the particles is significance or not. How about add the standard deviations?
--> According to your suggestion, we have added the standard deviations of the size of the particles in Table 2.
P3-Particle: Dn = 460±80 nm
P4-Particle: Dn = 550±110 nm
(5) Why there is no results for control experiments using P1 or P2 for preparation of particles and lectin binding assay? The reviewer thinks that is quite important to support the claims in this manuscript.
--> In order to prepare core-shell type polymer particles by dispersion polymerization in polar media, the use of an amphiphilic or hydrophilic macromonomer is generally required because it acts as a steric stabilizer during the process of particle formation. In this study, we have designed novel amphiphilic block copolymer-type macromonomers (P3 and P4) having pendant maltose residues and employed for the polymer particle preparation by dispersion polymerization in EtOH/H2O. The maltose-appended amphiphilic macromonomers were synthesized by CuAAC click reaction of the precursor macromonomers (P1 and P2) having pendant isobutyl and alkynyl groups. Compared to the favorable solubility of the amphiphilic macromonomers (P3 and P4) in EtOH/H2O (4/1, v/v), their precursors P1 and P2 were sparingly soluble in such polar media. That is, it would be impossible to prepare core-shell type polymer particles by employing P1 and P2 because they cannot stabilize the dispersion state. For better understanding, we added new sentences as follows;
“The occurrence of CuAAC click reaction with Mal-N3 led to the drastic change in solubility characteristics of the macromonomers. As a result, the obtained maltose-appended macromonomers (P3 and P4) were readily soluble in a mixture of EtOH and water, while the precursor macromonomers (MA-PIBVE-b-PVEEP, P1 and P2) were sparingly soluble in EtOH and insoluble in water.”
Concerning with this matter, we here describe an additional experimental result (not yet published). According to the similar manner described in this article, we have synthesized polyVE-based amphiphilic macromonomer having both a terminal methacryloyl function and pendant diethylene glycol methyl ether moieties (-OCH2CH2OCH2CH2OCH3). This amphiphilic macromonomer was capable of yielding core-shell type polymer particles by dispersion copolymerization with styrene in EtOH/H2O, and the resultant polymer particles afforded a stable aqueous dispersion. However, upon addition of Con A solution to the suspension, no Con A-triggered aggregation was observed, indicating the polymer particles in the present study are decorated by the maltose residues originating from the amphiphilic macromonomers [MA-PIBVE-b-P(VE-Mal), P3 and P4].
(6) Heating at 80 oC under acidic conditions for polymer particles for 3 h is a quite tough situation for polymers. The reviewer thinks that the poly(vinyl ether) components, especially near the acetal moieties, may be decomposed. In that case, the color of solutions may be changed, which may affect the absorbance values. Did the authors consider that possibility ?
--> Thank you for your valuable comments. After the hydrolysis of the maltose-decorated polymer particles at 80 oC under acidic conditions for 3 h, the reaction product maintained white color. This may be because the reaction mixture was obtained in a suspension of inherently white polymer particles. As you pointed, it may be reasonably acceptable that the polyVE component would decompose under such severe conditions. However, if some decomposition of the polyVE moiety occurred, we think it does not affect on the results of the quantitative estimation of the maltose residues on the particle surfaces by anthrone test. As you know, anthrone test was performed to evaluate the amount of green color complex made from anthrone and a furfural derivative originating from the liberated glucose by using colorimetry at 620nm. Since the decomposed polyVE moieties, even if they formed, does not possess any significant absorption in this region, accuracy of the quantitative analysis is not impaired.
Round 2
Reviewer 2 Report
The answers from authors are very carefully explained one by one. Also, I understood some difficulties in the experimental conditions. I think that the revised manuscript is improved in terms of some points raised at the previous peer-review. I am still worried about decomposition of the sample during the hydrolysis, however, as authors explained, it may not an essential point in this work.